# Diversity and Genetic Structure of *Dioon holmgrenii* (Cycadales: Zamiaceae) in the Mexican Pacific Coast Biogeographic Province: Implications for Conservation

**DOI:** 10.3390/plants10112250

**Published:** 2021-10-21

**Authors:** Mario Valerio Velasco-García, Carlos Ramírez-Herrera, Javier López-Upton, Juan Ignacio Valdez-Hernández, Higinio López-Sánchez, Lauro López-Mata

**Affiliations:** 1Centro Nacional de Investigación Disciplinaria en Conservación y Mejoramiento de Ecosistemas Forestales-Instituto Nacional de Investigaciones Agrícolas Pecuarias y Forestales (INIFAP), Avenida Progreso 5, Coyoacán, Ciudad de Mexico 04010, Mexico; taxodium01@hotmail.com; 2Colegio de Postgraduados, Carretera Mexico-Texcoco km 36.5, Montecillo, Texcoco 56230, Mexico; jlopezupton@gmail.com (J.L.-U.); ignaciov@colpos.mx (J.I.V.-H.); laurolopezmata@gmail.com (L.L.-M.); 3Colegio de Postgraduados, Boulevard Forjadores de Puebla No. 205, Santiago Momoxpan, San Pedro Cholula. C.P., Puebla 72760, Mexico; higiniols@colpos.mx

**Keywords:** heterozygosity, fixation index, genetic distance, endemic, conservation

## Abstract

*Dioon holmgrenii* De Luca, Sabato et Vázq.Torres is an endangered species; it is endemic and its distribution is restricted to the biogeographic province of the Mexican Pacific Coast. The aim of this work was to determine the diversity and genetic structure of nine populations. The genetic diversity parameters and Wright’s F statistics were determined with six microsatellite loci. The genetic structure was determined by using the *Structure* software and by a discriminant analysis. The genetic diversity of the populations was high. The proportion of polymorphic loci was 0.89, the observed heterogeneity was higher (Ho = 0.62 to 0.98) than expected (He = 0.48 to 0.78), and the fixation index was negative (IF = −0.091 to −0.601). Heterozygous deficiency (FIT = 0.071) was found at the species level and heterozygotes excess (FIS = −0.287) at the population level. The genetic differentiation between populations was high (FST = 0.287), with the number of migrants less than one. Three groups of populations were differentiated, and the variation within populations, between populations, and between groups was: 65.5, 26.3, and 8.2%, respectively. Multiple factors explain the high genetic diversity, while the genetic structure is due to geographic barriers. Community reserves are urgent in at least one most diverse population of each group.

## 1. Introduction

Cycads are gymnosperms that originated possibly in the Carboniferous and Permian periods of the Paleozoic era, between 345 to 280 million years ago [1,2,3], while living cycads originated during the Cenozoic era [4,5,6]. This group of plants has an evolutionary importance due to their ancestral origin, and they are considered the oldest lineage of seed plants. In addition, they maintain specific interactions with organisms for seed dispersal, pollination, for obtaining nitrogen and water, and they constitute the only food for some species of butterflies [7,8,9,10,11].

Currently, there are two families—Cycadaceae and Zamiaceae—, 10 genera, and 356 species in the world, distributed mainly in subtropical areas and, to a lesser extent in equatorial habitats with low temperature and humidity [12,13]. Mexico is the center of diversity for the Zamiaceae family with three genera and 63 species [13]. This represents 17.7% of the diversity of the Cycads order [13]. The 92% of these species are endemic to Mexico and 79% are threatened [14], but according to the IUCN nomenclature, all of them are at risk [13].

*Dioon* is a neotropical genus with 16 species, with 15 of them endemic to Mexico and one to Honduras; except for the non-evaluated species, all of them are threatened [13,14] mainly due to changes in land use. Four clades of the *Dioon* genus are recognized [15,16,17], all of them with a common ancestor in eastern Mexico, one clade expanded and diversified into southeastern Mexico and Honduras (Clado Spinulosum). Another clade diversified into three lineages that spread to the northeast (Clado Tomasellii), south (clade Purpusii), and northwest (clade Edule) of Mexico [17]. Between the Miocene and the Pleistocene epoch, the diversification and expansion of the *Dioon* genus was driven by habitat change, from humid to arid, caused by orogenic events and climate change [15,16,17]. Moreover, the biogeographic provinces could have provided ecological conditions that facilitated the speciation of the *Dioon* genus [17].

*Dioon holmgrenii* De Luca, Sabato & Vázq. Torres is endemic to Mexico, and it is distributed in the biogeographic province (sensu [18]) of the Mexican Pacific Coast [19,20,21]) on the border with the biogeographic province Sierra Madre del Sur (SMS), where some populations may exist [17]. The change of land use for rain-fed agriculture and intensive cattle raising decreases the density of individuals, modifies the population structure and the spatial dispersion of the individuals of *D. holmgrenii* [20]; therefore, it is listed as endangered species in the appendix II of CITES, in the Mexican NOM-059-SEMARNAT-2010 and by the IUCN [14,22,23].

Evolutionary factors define the genetics of species, populations; and individuals; these factors are in turn influenced by historical events and contemporary factors [24,25,26]. Natural selection, genetic drift, mating system, mutation, and gene flow stimulate local adaptations and lead to the genetic differentiation of populations [27,28]. The evaluation of diversity and genetic structure allows to identify exogamy, endogamy, gene flow, and deleterious mutation problems [29]; it also allows to evaluate the effect of habitat management and disturbances, viability, evolutionary potential, and to estimate the threat degree of populations and species [29,30,31,32], which in turn, allows the design of plans for their conservation [29,33,34].

The genetic variation of species is related to the size of their populations [29]. Relict and endangered populations have less genetic variation than large and non-threatened populations [29,30,35]. Therefore, low levels of diversity and genetic structure would be expected for the *Dioon* genus due to its endangered status and endemism [13,14]. However, *D. edule* Lind., *D. angustifolium* Miq., *D. sonorense* (De Luca, Sabato et Vázq.Torres) Chemnick, T.J. Greg. & Salas-Mor., *D. tomasellii* De Luca, Sabato & Vázq.Torres, *D. merolae* De Luca, Sabato et Vázq.Torres, *D. caputoi* De Luca, Sabato & Vázq.Torres, and *D. spinulosum* Dyer ex Eichler [26,36,37,38,39,40,41] show high genetic diversity. The high genetic diversity of rare plants may be due to the recent reduction of their populations and to the recurrent and excessive flow of genes [42]. The magnitude and distribution of the genetic diversity of species of the *Dioon* genus is defined by its life and biogeographic history related to the Pleistocene glaciations, recent local ecological factors, and gene flow between populations [26,41,43].

Dominant markers revealed low genetic diversity of two *D. holmgrenii* populations (Rancho el Limón and San Bartolomé) compared to two species from the Pacific Coast [38] and other species from the Purpusii clade [39,41]. However, SSR markers showed high genetic diversity of the Rancho el Limón and San Bartolomé populations [44]. According to this, only the genetic diversity of two populations have been evaluated, out of 10 populations that have been reported [20,21] in a small portion of the Mexican Pacific Coast biogeographic province. Knowledge of the diversity and genetic structure of *D. holmgrenii* populations is necessary to design the necessary conservation plans because of its endangered condition [13,14]). Therefore, the aim of this study was to determine the level and differences in genetic diversity between populations, as well as to know the genetic structure of nine populations of *D. holmgrenii* with microsatellite markers.

## 2. Results

### 2.1. Genetic Diversity

In the *D. holmgrenii* populations, 66 alleles were found in the six loci included in this study. Additionally, 20 to 39 alleles were found in the populations, where 18 to 37 were common, 2 to 7 were rare, and only in four populations there were 1 to 4 exclusive alleles (Table 1).

All loci were polymorphic in four populations, and in five, at least one locus was monomorphic; however, the level of polymorphism was high per population since the percentage of polymorphic loci ranged from 67 to 100% (Table 1). Statistical differences between populations were found for the number of alleles per locus (*p* = 0.0035), effective alleles per locus (*p* = 0.0062), expected heterozygosity (*p* = 0.0062), observed heterozygosity (*p* = 0.0123), and fixation index (*p* = 0.004). The Ocotlán population showed the highest values of genetic diversity, opposing to the Rancho Viejo population, which resulted in the lowest values, except for the observed heterozygosity, where the La Lima population had the highest value and the Rancho el Limón and Cerro Antiguo populations presented the lowest values (Table 1). The Rancho el Limón population also had the lowest values for the number of alleles per locus and observed heterozygosity. The fixation index was negative in all populations, with less fixation in La Lima (Table 1).

### 2.2. Genetic Structure

The mean value of the total fixation index (F_IT_) was positive and low. Half of the loci had negative values, and the Zam29 locus had the highest value (Table 2). The fixation index at population level (F_IS_) was negative for all loci, except for the Zam29 locus (Table 2). The fixation index between populations (F_ST_) showed a higher genetic diversity (72.2%) inside populations than between them (27.8%, F_ST_ = 0.278) (Table 2); and lastly, the Ed6 locus showed the highest differentiation between populations.

The number of migrants per generation (Nm) was lower than one for the five loci (Table 2); whereas more than one migrant per generation was found in one locus (Table 2). Nei’s genetic distances and F_ST_ values between pairs of populations showed that the lowest genetic differentiation was found between the Cerro Caballo and San Bartolomé populations, and between Rancho el Limón and Cieneguilla, while the highest genetic differentiation occurred between Río Leche and La Lima with Rancho Viejo, Rancho El Limón, and Cieneguilla (Appendix A). The genetic distance between populations had a positive and significant correlation (r = 0.6822, *p* = 0.0040) with the geographic distance of the populations (Figure 1).

The number of groups suggested by the Structure Harvester software was equal to three (K = 3), followed by four (K = 4) (Figure 2). For K equal to 3, a clear genetic differentiation was observed between populations; group one was made up of the Río Leche, La Lima, and Ocotlán towns; group two included Rancho Viejo, Rancho el Limón, and Cieneguilla; and group three consisted of the Cerro Antiguo, Cerro Caballo, and San Bartolomé populations (Figure 2). The previous result was consistent with the result of the Discriminant Analysis of Principal Components (DAPC) (Figure 3). In the ADCP, 96.8% of the total variance was explained by 40 principal components. The analysis of molecular variance, considering the groups (K = 3) suggested by the Structure software, showed that 65.54% of the genetic variation is found within populations, 26.25% between populations, and 8.21% between groups (Table 3).

## 3. Discussion

### 3.1. Genetic Diversity

Generally, genetic diversity in plant species with small populations is reduced as a consequence of genetic drift and inbreeding [45,46,47]. However, in this study, it was found that the level of genetic diversity in *D. holmgrenii* was high and superior (Table 1) to the values of genetic diversity found in endemic and woody perennial species [48,49], despite the fact that *D. holmgrenii* usually grows in fragmented and isolated populations as a consequence of the change in land use for rainfed agriculture and livestock [20,21]. These observations confirmed that rare species conserve high levels of genetic diversity [50,51].

The results of the present study showed that *D. holmgrenii* populations have high levels of genetic diversity (Table 1) compared to *D. edule*, *D. angustifolium*, and *D. sonorense* [37,44,52] evaluated with SSR and ISSR markers. Likewise, the Ho of the *D. holmgrenii* populations was higher than that reported in several species of the Zamia genus evaluated with SSR markers [53,54,55,56,57,58,59,60,61]. On the other hand, the genetic diversity values were higher in comparison with other species of the Dioon *genus* evaluated with dominant markers [26,36,38,39,40]. The latter was expected because it has been shown that SSR markers show higher values of genetic diversity than dominant markers; for example, the genetic diversity values of the Rancho el Limón and San Bartolomé populations evaluated with SSR markers were higher (Ho = 0.677, He = 0.605, NA = 1.6, Table 4) [44] compared to enzymatic markers (Ho = 0.204, He = 0.170, NA = 1.71, Table 4) [38]. Among the species of the Zamiaceae family evaluated with SSR markers, only *D. edule* [44] showed levels of genetic diversity similar to those in the *D. holmgrenii* populations considered in this study (Table 1); the He and the number of alleles per locus of *D. edule* was higher than all the populations of *D. holmgrenii* (except Río Leche and Ocotlán), while the Ho of *D. edule* was higher than in Rancho Viejo, Rancho el Limón, Cerro Antiguo, and Cieneguilla populations.

The high level of genetic diversity observed in this study (Table 1) partially agreed with a previous evaluation of individuals from the Rancho el Limón and San Bartolomé populations using different SSR markers than those used in this research [44]. Compared to another study [44], all populations (except Rancho Viejo) had higher values of NA, He, and Ho, respectively; while, San Bartolomé had Ho similar to that reported by Prado [44].

In all *D. holmgrenii* populations, Ho was higher than He (Table 1). This pattern was also observed for the Rancho el Limón and San Bartolomé *D. holmgrenii* populations in previous studies, both with SSR (Ho = 0.677, He = 0.605, Table 4) [44] and enzymatic markers (Ho = 0.204, He = 0.170, Table 4) [38]. In addition, this pattern is common in *Dioon* species, such as *D. edule* [26,37], *D. caputoi* [39], *D. sonorense*, *D. tomasellii* [38], *D. merolae* [41], and *D. angustifolium* [44], as well as in the *Zamia pumila* L. complex [54] (Table 4). The fact that He was higher than Ho in *D. holmgrenii* was congruent with the excess of heterozygous individuals shown by the negative values of the fixation index (F) (Table 1), which means that the possible effect of gene drift and inbreeding is counteracted by natural selection that favors the highest percentage of heterozygous individuals who can adapt to environmental changes [26]. Therefore, it can be assumed that the *D. holmgrenii* populations are in good genetic conditions. 

The high genetic diversity in this species may be a consequence of the combined effect of factors such as: longevity of the individuals, interbreeding system, the extensive geographical distribution, the variety of environments it occupies, and the evolutionary history that it shares with other species of the *Dioon* genus [38,39,41]. The longevity of the *D. holmgrenii* individuals is unknown; however, in the populations under study, there were individuals of up to 6.5 m tall [20,21], which might be around 3892 years old, if the growth rate was similar to the average growth rate of *D. edule* [62]. The longevity of this species together with its phenotypic plasticity might be the reasons of its survival for several decades under adverse environmental conditions, which is the time when recombination can produce better adapted genotypes [49]. On the other hand, the *D. holmgrenii* dioecious breeding system is a mechanism that prevents self-fertilization, decreasing the probability of inbreeding depression and keeping high genetic loads due to higher mutation rates per generation [63,64]. 

The area occupied by the populations evaluated in this study was 4197.33 ha (Table 5); however, the distribution is broader, considering the Llano de León population of 620 ha [20,21], as well as the Jamiltepec and Juchatenco populations (which extensions are unknown) [15,17]. In addition to the above, other factors contributing to the high genetic diversity might be the wide ranges of elevation and variations of the soil compositions, climate, and vegetation (Table 5) where *D. holmgrenii* is distributed.

### 3.2. Genetic Structure

In this study, half of the polymorphic loci of *D. holmgrenii* were excessively heterozygous (negative F_IT_) (Table 2), and the other half were deficient in heterozygous individuals [65,66,67]. However, average F_IT_ showed a slight heterozygous deficiency (Table 2). Similarly, higher deficiencies of heterozygotes (F_IT_ = 0.116 a 0.336) have been reported for *D. angustifolium*, *D. sonorense*, *D. tomasellii,* and *D. edule* [36,38,40]. On the contrary, in *D. edule*, *D. caputoi,* and *D. merolae* populations, an excess of heterozygotes was found (F_IT_ = −0.482 a −0.172) [26,39,41].

The negative value of the average fixation index at the population level (F_IS_) of *D. holmgrenii*, showed an excess of heterozygous individuals (Table 2). Excess heterozygous individuals have been found for *D. edule*, *D. angustifolium, D. sonorense*, *D. tomasellii, D. caputoi,* and *D. merolae* (F_IS_ = −0.035 to −0.592, Table 4) [26,36,38,39,41]. In contrast, *D. edule* were heterozygous deficient (F_IS_ = 0.173, Table 4) [40]. Contrary to the results of this study, a heterozygote excess at population and total levels (F_IS_ = −0.201, F_IT_ = −0.116, Table 4) was reported in the *D. holmgrenii* populations from Rancho el Limón and San Bartolomé in a previous study [38].

The mean F_ST_ value showed a very high differentiation between populations [29,74] of *D. holmgrenii*, and it was higher with respect to species of the *Dioon* genus (F_ST_ = 0.060 a 0.194, Table 4) [26,36,38,39,40,41], *Zamia* (F_ST_ = 0.066 to 0.175, Table 4) [56,57,59,60,61], and *Ceratozamia* (F_ST_ = 0.047 to 0.141, Table 4) [75,76]. High differentiation between *D. holmgrenii* populations is the result of low gene flow [67], due to restrictions in the dispersion of seeds and pollen, which generally promote differentiation in species pollinated by insects and with seeds dispersed by gravity [77]. In agreement with the above, the average number of migrants per generation (Nm) of the *D. holmgrenii* populations was low (Table 2) and less than the minimum value required to avoid genetic differentiation due to genetic drift [65,67]. The low Nm implies that the reduced genetic exchange in the *D. holmgrenii* populations caused high differentiation between them, which may be associated with fragmentation, isolation, and the small size of most populations (Table 5) [20]. As for the high differentiation and the low number of migrants between *D. holmgrenii* populations, the average genetic distance was high and higher than that reported for populations of *D. edule* (0.04) [26], *Cycas guizhouensis* K.M. Lan & R. F. Zou (0.054) [78], *Zamia amblyphyllidia* D.W. Stev. (0.206), and *Z. portoricensis* (Jacq.) H.M. Hern. (0.897) [56]. The positive relationship between genetic and geographic distances of *D. holmgrenii* populations may be the result of low gene flow between distant populations [26].

The topography of the distribution area of *D. holmgrenii* consists of depressions and elevations with significant unevenness (Figure 4), which constitute physical barriers that can limit gene flow and increase differentiation, generating allopatric populations [79,80]. In accordance with the above, the group one populations are clearly located west of the Río Verde, while the populations of group two and three are located east of Río Verde (Figure 2, Figure 3 and Figure 4). The depression formed by the Rio Verde flow and the mountains may be physical barriers that influence gene flow between groups. Moreover, the environmental variations promote divergence to more serum lineages in response to aridity [16]; in this sense, the populations of the west of the Rio Verde correspond to the more humid climate of the subhumid ones, while the Populations of the eastern Río Verde correspond to the climate of medium humidity and drier of the subhumid ones (Table 5).

On the other hand, clustering between *D. holmgrenii* populations can be explained by the hypothesis of precursor and derived populations [26]. In this case, group one (west of Río Verde) are precursors based on their high genetic diversity; while groups two and three (east of Río Verde) are derived populations for having less genetic diversity compared to west of Río Verde populations (Figure 2 and Figure 3). The closer proximity of the western populations of the Río Verde with the region with the highest diversity of species of the *Dioon* genus (Tehucacán-Cuicatlán valley surroundings), as well as their proximity to the distribution area of *D. caputoi* and *D. planifolium* that mixed with *D. holmgrenii* in the late Pleistocene era [15], supports the hypothesis of precursor populations.

The distribution of diversity among *D. holmgrenii* populations can be related to life history and biogeographic history [26,43]. In addition to this, the biodiversity in the region where this species is distributed is defined by the physiography and composition of its substrate, which are the result of a complex geological evolution [81]. In this sense, the current known distribution of *D. holmgrenii* is restricted to the Xolapa tectonostratigraphic terranes. The Xolapa terrain originated between the Late Cretaceous and Paleocene (99 to 54 Million years), but joined the Mixtec and Zapotec tectonic terrains between the Eocene and Middle Miocene (54 to 16 million years) [81]. Although extinct cycad genus already existed in the Mixtec terrain in the Middle Jurassic (180 to 159 million years) [82,83,84], the *Dioon* genus originated in the late Paleocene and early Eocene (~56 million years) in the north of the American continent and migrated to the south [6,16,85]. Through the migration process, the genus *Dioon* expanded throughout Mexico, where the change from humid to xeric habitat caused the greatest speciation during the Oligocene and late Miocene [4,16]) and possibly until the Pleistocene (2 to 0.5 million years) [15]. The southern clade or Purpusi [15,17], to which *D. holmgrenii* belongs, expanded towards the Pacific coast and it diversified due to the aridification process that occurred during the Miocene (23 to 5 million years) [16,17]. Reconstruction of the expansion of the Purpusi clade indicates that *D. holmgrenii* has as ancestors to *D. planifolium* Salas-Mor., Chemnick & T.J. Greg., *D. argenteum* T.J. Greg., Chemnick, Salas-Mor. & Vovides, *D. purpusii*, *D. caputoi,* and *D. califanoi* De Luca & Sabato [17], all of these currently distributed in Mixtec, Zapotec, and Juárez lands. In accordance with the above, *Dioon* first populated the Mixtec and Zapotec lands and later the union of the Xolapa terrain with these lands; the *Dioon* genus dispersed in the Xolapa terrain, where the speciation of *D. holmgrenii* possibly occurred, to the limit of the Mixtec land. All the above indicates that colonization of the *Dioon* genus occurred from west to east, and this fact reinforces the hypothesis of precursor (west of Río Verde) and derived (east of Río Verde) populations of *D. holmgrenii*.

Possibly, the recent colonization of the eastern Río Verde populations and a slight associated founder effect [26], could explain the lower genetic variation. Among the populations east of Río Verde, the group 3 (Rancho Viejo, Rancho el Limón and Cieneguilla) had the lowest genetic diversity, possibly because they are more fragmented and they have a greater deforested area (Table 5), with the presence of intensive and extensive livestock [20]. Habitat fragmentation generally reduces the size of plant populations and increases their isolation, leading to genetic erosion and increased genetic differentiation between populations [40,75].

### 3.3. Implications and Strategies for Conservation

*Dioon holmgrenii* has evolutionary potential to adapt to changing climate conditions due to its high genetic diversity, which allows its sustainable management [39]. The results suggest that inbreeding does not represent an immediate danger for the species, but genetic drift could have an effect in populations with a small number of individuals, such as Cerro Antiguo, Río Leche, and Rancho El Limón [21]. Likewise, the low flow of genes and the very high differentiation between populations suggests that the loss of some population, due to changes in the current land use [20] or meteorological phenomena, would imply the significant loss of genetic diversity [49].

This implies to include all or most of the populations in the conservation plans. The distribution of the genetic diversity into three groups allows to guide conservation efforts and at least one population from each group should be integrated into an in situ conservation program. Sites with the presence of cycads must be declared or included in the Natural Protected Areas (NPA) lists [86,87,88]. NPA are the main in situ conservation strategy in Mexico; however, NPA do not always coincide with sites where there are threatened species and they are not efficient in protecting these species [89,90]. The effective conservation of biodiversity may occur through the sustainable use of resources by the owners; therefore, community reserves and Management Units for the Conservation and Sustainable Use of Wildlife (UMA) are options to conserve *D. holmgrenii* populations [52,88,90].

Based on the genetic diversity parameters determined in this study, as well as on the surface, variation of environments, and level of conservation (Table 5) [20,21] of the populations that conform each group, at least the Ocotlán, Cieneguilla, and Cerro Caballo populations must be declared community reserves. At the second priority level, the Río Leche, Cerro Antiguo, and Rancho el Limón populations of groups one, two, and three, respectively, should be considered in in-situ conservation programs due to the presence of exclusive alleles. Likewise, in all *D. holmgrenii* populations it is necessary to implement UMAs aimed at reproducing, marketing, and planting plants in deforested sites. Through the UMAs, the owners or possessors can obtain benefits to avoid livestock and agriculture in the lands with the presence of this species. The foregoing is of vital importance in populations of the group three, where habitat fragmentation due to land use change is greater (Table 5) [20,21] and genetic diversity decreases, which may decrease the adaptive and evolutionary potential.

The owners of the San Bartolomé and Ocotlán populations have processed the authorization of UMAs at the Secretariat of the Environment and Natural Resources (SEMARNAT, México City, Mexico). However, the process has been delayed for several years, under the argument that the species is listed as “in danger of extinction” in the NOM-059-SEMARNAT-2010 [14] and in the IUCN red list [22]. This argument is wrong, because the UMAs are the strategy considered in the General Wildlife Law for the conservation and sustainable use of the species listed in NOM-059-SEMARNAT 2010. SEMARNAT should promote and prioritize the authorization of UMAs for *D. holmgrenii*; otherwise, the owners of the land, not having tangible benefits from the species nor a way of subsistence, will continue to eliminate the natural vegetation to allocate the land to rainfed agriculture and intensive livestock [20].

## 4. Materials and Methods

### 4.1. DNA Sampling and Extraction

Leaflets of between 30 to 36 individuals (285 in total) were collected in nine populations of *D. holmgrenii* (Figure 4) located in the bio-geographic province Costa del Pacífico Mexicano, in the state of Oaxaca, Mexico. The sample included all stages of development in each population [20], as well as the extension and the altitudinal variation (Table 5). The minimum distance between sampled plants was 100 m in all populations, except in the Rancho el Limón population (minimum distance 30 m) due to its size (˂5 ha). Leaflet samples were placed in plastic bags and placed on ice to be transported to the laboratory, where they were stored at −20 °C until DNA extraction.

DNA extraction was performed on 100 mg of tissue with the commercial ChargeSwitch^®^ gDNA Plant DNA extraction kit from Invitrogen (Carlsbad, CA, USA), and using a DNA extraction robot (King Fisher Flex, Thermo Scientific^TM^, Wilmington, DE, USA). The quality (absorbance at 260/280 nm) and the DNA concentration were measured with an ultra-low volume spectrometer (NanoDrop 2000 UV-Vis Spectrophotometer, Thermo Scientific^TM^, Wilmington, DE, USA).

### 4.2. Microsatellite Analysis

Six nuclear microsatellite primers developed for *D. edule* [36] and 10 developed for *Zamia integrifolia* L.f. [53] were tested, and the amplification of six of them were used in this study (Appendix A). Genomic DNA was diluted to a concentration of 20 ng µL^−1^ with HPLC water. PCR (polymerase chain reaction) was performed in a thermal cycler (C1000TM Thermal Cycler, Bio Rad^®^, Hercules, CA, USA). All PCR products were verified on 2% agarose gels. Denaturation, alignment, and extension temperatures were adjusted according to the results observed in the agarose gels.

The microsatellite amplification of all the samples was carried out by multiplex PCR, where the primers with similar alignment temperatures and with different size (bp) were grouped into three groups (Appendix A). The primers were labeled with fluorescent labels at the 5’ end (Appendix A) for their detection in a fragment sequencer. Each 25-μL multiplex PCR reaction mix consisted of 5 μL of 5X Buffer PCR, 0.6 μL of 25 mM MgCl_2_, 0.5 μL of 10 mM DNTP, 5 μL of 10 pM of each forward primer pair (2.5 μL each), 5 μL of 10 pM of each reverse primer pair (2.5 μL of each), 0.25 μL of 5 units of Taq DNA polymerase (GoTaq Flexi DNA Polymerase, Promega) and 2.0 μL of template DNA (20 ng μL^−1^) and 6.65 μL of HPLC water. The multiplex PCR program consisted of: 5 min of initial denaturation at 94 °C; followed by 35 cycles of 30 s of denaturation at 94 °C; one minute of alignment at 58.1 (group 1), 57.3 (group 2) or 58.9 °C (group 3); 2 min extension at 72 °C, with a 15 min final extension at 72 °C.

The multiplex PCR products were denatured at 96 °C for three minutes in a thermal cycler (C1000TM Thermal Cycler, Bio Rad^®^, Hercules, CA, USA) and then frozen at −20 °C for three minutes before starting electrophoresis. The electrophoresis of the PCR products was carried out on a DNA sequencer (Genetic Analyzer 3130, Applied Biosystems, Foster City, CA, USA). Each electrophoresis reaction consisted of 2 μL of PCR product, 0.25 μL of Size Standard LIZ^®^500, and 7.75 μL of formamide. The electropherograms were analyzed with the GeneMapper^®^ software version 4.0 (Foster City, CA, USA) [91], to construct a matrix with the genotypes of each individual.

### 4.3. Genetic Diversity

Genetic diversity parameters such as the percentage of polymorphic loci (PLP), average number of alleles (NA), effective number of alleles (EA), expected heterozygosity (He), observed heterozygosity (Ho), and fixation index (F) for each population were determined with the InfoGene software version 2016 [92]. A variance analysis and multiple comparisons of means were performed with the non-parametric Friedman test [93], in order to know the differences between the populations in NA, EA, He, Ho, and F.

The numbers of total (NTA) common (NCA), rare (NRA), and exclusive (NAE) alleles were identified for each population. Alleles were called rare when their frequency was less than 0.05, and common when their frequency was equal to or higher than 0.05 [94]. The exclusive alleles were those that were only found in one population.

### 4.4. Genetic Structure

The F statistics [95] were estimated in order to know the fixation of alleles or the effects of inbreeding at the population level (F_IS_), between populations (F_ST_) and considering all populations as one (F_IT_). The number of migrants per generation (Nm) was calculated with Nm = (1 − F_ST_)/(4 F_ST_) [89]. Additionally, Nei’s genetic distances [96], geographic distances, and F_ST_ values between pairs of populations were calculated. The relationship between Nei’s genetic distances and geographic distances was determined with the Mantel test [97] using the Vegan 2.5–6 package of the R version 4.00 statistical software [98].

In order to evaluate the genetic structure between populations, the Structure V.2.4 software was used [99]. This computer program assigns the individuals to different genetic groups (K), without considering the source population by using Bayesian probability and based on allele frequencies. Around 500,000 to 1,000,000 of Monte Carlo Markov chains were used, K values from 1 to 10, and 10 repetitions for each K. The results were run with Structure Harvester v 0.6.94 software [100]. Evanno’s test [101] was used to determine the K value that fit the data. Additionally, in order to identify the number of groups, the discriminant analysis of principal components was performed with the adegenet library [102,103] of the statistical program R version 4.0.0 [98]. An AMOVA analysis of molecular variance [104] was done considering the groups obtained from Structure, with the InfoGen software [86].

## 5. Conclusions

*Dioon holmgrenii* has a high genetic diversity. The high diversity of *D. holmgrenii* is the result of the combined effect of its interbreeding system, longevity, variety, and relative extensive geographic distribution, as well as the evolutionary history that it shares with other species. The genetic structure is defined by its evolutionary history and biogeographic history; likewise, the Río Verde and the mountains act as a geographical barrier for the differentiation of population groups. Knowledge of diversity and genetic structure allows creating the in situ conservation strategies for *D. holmgrenii*.

## Figures and Tables

**Figure 1 plants-10-02250-f001:**
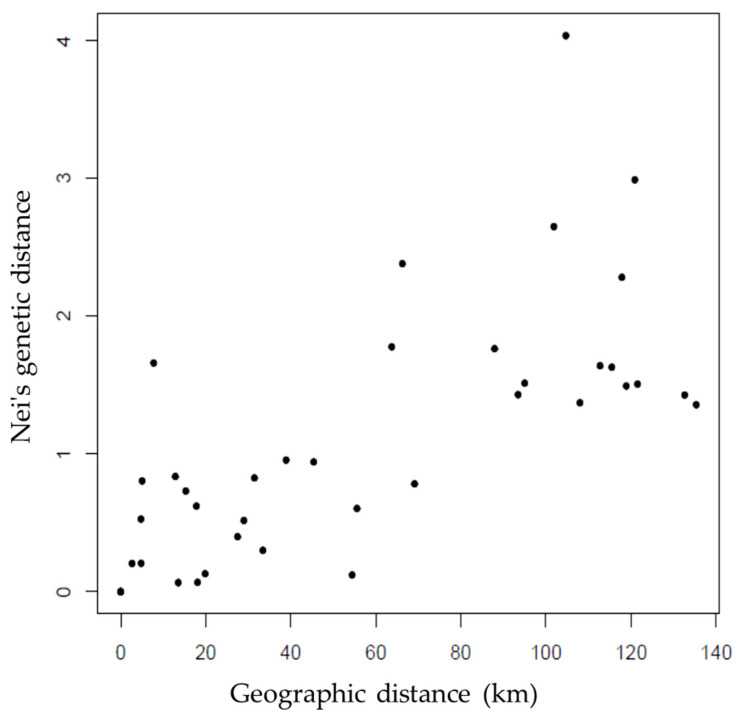
Relationship between genetic distances and geographic distances for *Dioon holmgrenii* populations.

**Figure 2 plants-10-02250-f002:**
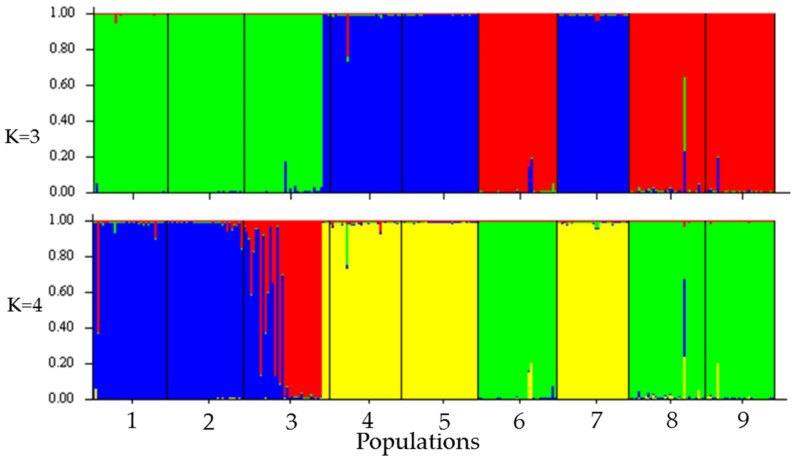
Bar diagram of the Structure analysis, showing three (K = 3) and four (K = 4) groups of populations (1 = Río Leche, 2 = La Lima, 3 = Ocotlán, 4 = Rancho Viejo, 5 = Rancho el Limón, 6 = Cerro Antiguo, 7 = Cieneguilla, 8 = Cerro Caballo, and 9 = San Bartolomé) of *Dioon holmgrenii*.

**Figure 3 plants-10-02250-f003:**
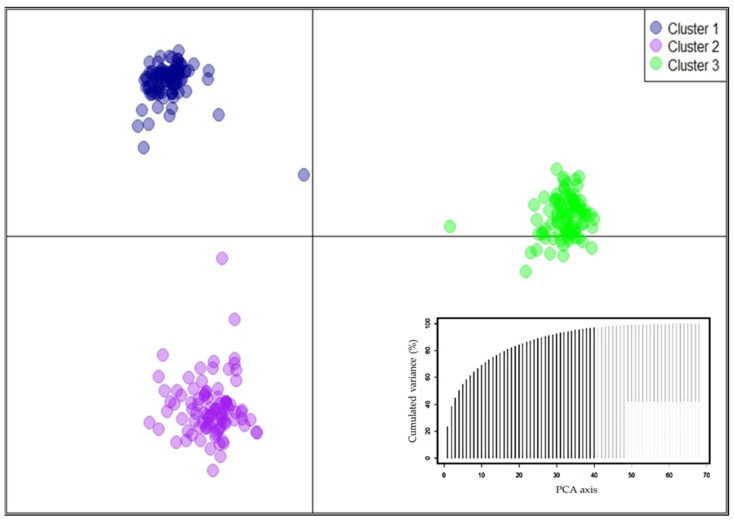
Grouping of individuals of *D. holmgrenii* through the discriminant analysis of principal components.

**Figure 4 plants-10-02250-f004:**
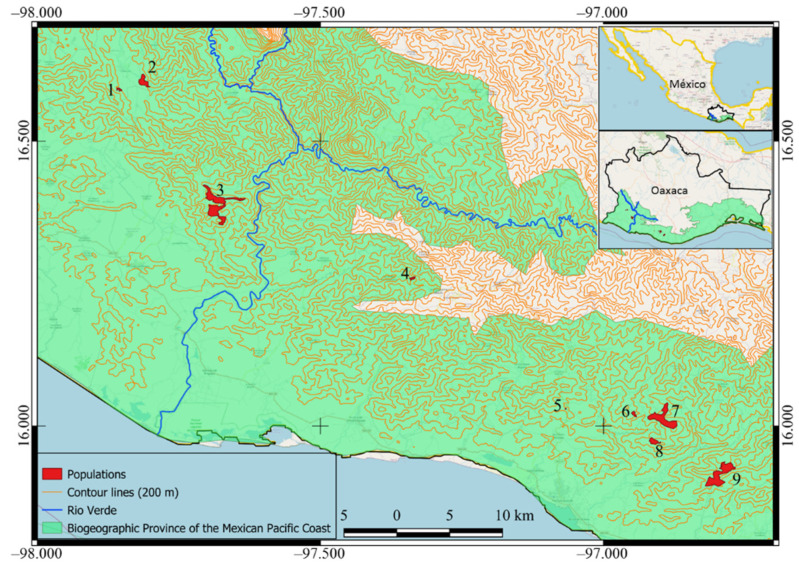
Location of nine populations of *Dioon holmgrenii* (1 = Río Leche, 2 = La Lima, 3 = Ocotlán, 4 = Rancho Viejo, 5 = Rancho El Limón, 6 = Cerro Antiguo, 7 = Cieneguilla, 8 = Cerro Caballo, 9 = San Bartolomé) in the biogeographic province of the Mexican Pacific Coast, in Oaxaca, Mexico.

**Table 1 plants-10-02250-t001:** Genetic diversity obtained with six nuclear microsatellites for nine *Dioon holmgrenii* populations. Different letters within one column denote statistically significant differences (*p* < 0.05) by Friedman test. NTA, number of total alleles; NCA, number of common alleles; NRA, number of rare alleles; NAE, number of exclusive alleles; PLP, percentage of polymorphic loci; NA, number of alleles per locus; EA, effective alleles per locus; He, expected heterozygosity, Ho, observed heterozygosity; F, fixation index.

Populations	NTA	NCA	NRA	NAE	PLP	NA		EA		He		Ho		F	
Río Leche	37	32	5	3	100	6.17	ab	4.02	ab	0.72	ab	0.97	ab	−0.352	Ab
La Lima	24	21	3	0	100	4.00	bc	2.89	bc	0.63	bc	0.98	a	−0.601	A
Ocotlán	39	37	2	1	100	6.50	a	4.76	a	0.78	a	0.95	ab	−0.231	Bc
Rancho Viejo	20	18	2	0	83	3.33	c	2.41	c	0.48	c	0.66	bc	−0.121	Bc
Rancho el Limón	22	18	4	1	83	3.67	c	2.55	bc	0.49	bc	0.62	c	−0.108	C
Cerro Antiguo	26	22	4	4	83	4.33	bc	3.46	bc	0.60	bc	0.63	c	−0.091	C
Cieneguilla	23	20	3	0	67	3.83	bc	3.07	bc	0.51	bc	0.64	bc	−0.209	Bc
Cerro Caballo	25	32	7	0	100	4.17	bc	2.55	bc	0.56	bc	0.74	abc	−0.299	Bc
San Bartolomé	37	21	4	0	83	4.00	bc	2.72	bc	0.56	bc	0.69	bc	−0.227	Bc

**Table 2 plants-10-02250-t002:** Values of Wright’s F statistics (total fixation index (F_IT_); fixation index at population level (F_IS_); The fixation index between populations (F_ST_)) and number of migrants per generation (Nm) for six loci in nine *Dioon holmgrenii* populations.

Locus	F_IT_	F_IS_	F_ST_	Nm
Ed3	−0.143	−0.526	0.251	0.746
Cap5	−0.095	−0.433	0.236	0.812
Ed5	0.009	−0.288	0.230	0.836
Ed6	0.296	−0.453	0.515	0.235
1660	−0.155	−0.310	0.118	1.865
Zam29	0.515	0.290	0.317	0.539
Average	0.071	−0.287	0.278	0.839

**Table 3 plants-10-02250-t003:** Molecular variance analysis (AMOVA) between groups, between populations, and within populations of *Dioon holmgrenii*.

Variation Source	Degrees of Freedom	Sum of Squares	Mean Squares	Probability	Variance Component	Percentage
Between groups	2	118.07	59.03	0.0700	0.53	8.21
Between populations	6	194.72	32.45	0.0001	1.7	26.25
Within populations	139	589.2	4.24	0.0001	4.24	65.54
Total	147	901.99	6.14		6.47	100

**Table 4 plants-10-02250-t004:** Summary of the genetic and genetic diversity of the genus *Dioon* and *Zamia*. NA, number of alleles per locus; PLP, percentage of polymorphic loci; PA, private alleles, Ho, observed heterozygosity, He, expected heterozygosity, F, fixation index, FIT, total fixation index; FIS, fixation index at population level; FST, fixation index between populations; Nm, migrants per generation; --, no data.

Species	Markers	Genetic Diversity						Genetic Structure				Reference
		NA	PLP	PA	Ho	He	F	F_IT_	F_IS_	F_ST_	Nm	
*Dioon angustifolium*	Allozyme	--	52.4	--	0.215	0.218	--	0.165	−0.007	0.167	1.55	[36]
*D. angustifolium*	SSR	2.86	--	--	0.519	0.478	--	--	--	--	--	[44]
*C. caputoi*	Allozyme	45.5	76.9	--	0.522	0.358	--	−0.354	−0.452	0.060	--	[41]
*D. caputoi*	Allozyme	1.91	78.95	--	0.490	0.450	--	−0.242	−0.379	0.099	--	[39]
*D. edule*	Allozyme	--	54.78	--	0.273	0.239	--	−0.172	−0.270	0.075	2.980	[26]
*D. edule*	Allozyme	2.09	95.41	--	0.323	0.386	0.274	0.336	0.173	0.194	--	[40]
*D. edule*	SSR	--	--	--	0.304	0.264	--	--	--	--	--	[37]
*D. edule*	SSR	5.86	--	--	0.664	0.714	--	--	--	--	--	[44]
*D. holmgrenii*	Allozyme	1.71	63.16	--	0.204	0.170	--	−0.116	−0.201	0.069	--	[38]
*D. homgrenii*	SSR	3.57	--	--	0.677	0.605	--	--	--	--	--	[44]
*D. merolae*	Allozyme	29.5	92.3	--	0.713	0.446	--	−0.482	−0.592	0.070	--	[41]
*D. spinulosum*	SSR	3.57	--	--	0.452	0.514	--	--	--	--	--	[44]
*D. sonorense*	Allozyme	2.00	81.58	--	0.330	0.314	--	0.130	−0.025	0.151	--	[38]
*D. sonorense*	ISRR	--	41.46	--	0.078	0.082	0.027	--	--	0.045	--	[52]
*D. tomasellii*	Allozyme	1.96	83.15	--	0.309	0.295	--	0.116	−0.035	0.145	--	[38]
*Zamia amblyphyllidia*	SSR	5.37	--	4.00	0.466	0.482	0.039	--	--	0.160	0.720	[56]
*Z. erosa*	SSR	5.55	100.00	24.00	0.545	0.549	0.008	--	--	--	0.095	[55]
*Z. incognita*	SSR	--	--	--	0.402	0.401	--	--	--	0.155	--	[60]
*Z. inermis*	SSR	5.95	70.84	--	0.212	0.402	0.473	--	--	0.734	0.091	[58]
*Z. integrifolia*	SSR	5.20	--		0.572	0.603	0.041	--	--	--	--	[53]
*Z. lacayana*	SSR	4.70	--	12.33	0.484	0.490	0.061	--	--	0.066	3.596	[57]
*Z. melanorrhachis*	SSR	--	--	--	0.264	0.284	--	--	--	--	--	[60]
*Z. pumila* Complex	SSR	8.27	97.33	--	0.489	0.520	0.071	--	--	0.137	1.780	[59]
*Z. pumila* L. Complex	SSR	4.00	75.60	6.90	0.398	0.385	0.040	--	--	0.175	--	[61]
*Z. pumila* Complex	SSR	5.11	100.00	11.80	0.625	0.436	0.307	--	--	--	--	[54]
*Z. pumila*	SSR	5.55	100.00	20.33	0.549	0.565	0.019	--	--	--	2.400	[55]
*Z. pumila*	SSR	6.00	--	0.00	0.454	0.477	0.015	--	--	--	--	[56]
*Z. portorensis*	SSR	6.30	100.00	18.33	0.517	0.521	−0.004	--	--	--	2.633	[55]
*Z. portoricensis*	SSR	5.67	--	1.67	0.450	0.446	0.007	--	--	0.038	2.650	[56]

**Table 5 plants-10-02250-t005:** Characteristics of the *Dioon holmgrenii* populations evaluated with SSR markers.

Populations	Latitude	Longitude	Elevation Range	Area (ha)	Deforested Area (%)	Soil Type [68,69]	Climate [70]	MAT (°C) [71]	MAR (mm) [72]	Vegetation Type [73]
Río Leche	16°35.252′ to 16°35.635′	−97°51.052′ to −97°51.583	940 to 1050	30.47	8.53	Dystric luvisol	Aw2	22 to 24	2000 to 2500	Pine-oak forest
La Lima	16°35.710′ to 16°37.076′	−97°40.072′ to −97°49.263	810 to 960	234.50	55.96	Chromic luvisol	Aw2	22 to 24	2000 to 2500	Pine-oak forest and tropical deciduous forest
Ocotlán	16°21.172′ to 16°25.415′	−97°37.967′ to −97°42.458	750 to 1310	1885.00	13.93	Chromic acrisol, eutric cambisol	Aw2	22 to 24	2000 to 2500	Oak forest and pine forest
Rancho Viejo	16°20.309′ to 16°15.355′	−97°19.962′ to −97°20.572	940 to 1400	35.20	38.64	Chromic luvisol	(A)C(w2), Aw2	20 to 22	2000 to 2500	Oak forest and pine-oak forest
Rancho El Limón	16°1.715′ to 16°1.873′	−97°3.936′ to −97°4.105′	590 to 650	5.00	56.60	Eutric regosol	Awo	24 to 26	2000 to 2500	Oak forest and tropical semi-deciduous forest
Cerro Antiguo	16°0.996′ to 16°1.628′	−96°56.506′ to −96°57.044′	790 to 980	48.20	19.33	Eutric regosol	Aw1	22 to 26	2000 to 2500	Oak forest
Cieneguilla	15°59.713′ to 16°2.468′	−96°52.206′ to −96°55.314′	650 to 1030	849.51	28.66	Eutric regosol	Aw1	24 to 26	1500 to 2500	Oak forest and pine-oak forest
Cerro Caballo	15°58.055′ to 16°58.801′	−96°53.940′ to −96°58.402′	560 to 1020	139.95	1.99	Eutric regosol	Aw1, Awo	24 to 26	1500 to 2000	Oak forest and pine-oak forest
San Bartolomé	15°53.637′ to 16°56.240′	−96°45.996′ to −96°49.235′	510 to 1010	969.50	8.73	Eutric regosol	Awo	22 to 26	1200 to 1500	Oak forest, tropical deciduous forest and pine-oak forest

MAT, mean annual temperature; MAR, mean annual rainfall; Awo, warm subhumid (low humidity) with summer rains, 5 to 10.2% winter rain, <60 mm of rainfall in the driest month, mean annual temperature >22 °C; Aw1, warm subhumid (intermediate humidity) with summer rains, 5 to 10.2% winter rain, <60 mm precipitation in the driest month, mean annual temperature >22 °C; Aw2, warm subhumid (high humidity) with summer rains, 5 to 10.2% winter rain, <60 mm precipitation in the driest month, mean annual temperature >22 °C; (A)C(w2), semi-warm temperate subhumid (high humidity) with rains in summer, 5 to 10.2% of winter rain, <60 mm of rainfall in the driest month, mean annual temperature of 18 to 22 °C.

## Data Availability

The data present in this study are available on request from the correspondent author. We will look for the procedure to make data available in a publicly accessible repository.

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
