# Peer review of "Diversity and Genetic Structure of Dioon holmgrenii (Cycadales: Zamiaceae) in the Mexican Pacific Coast Biogeographic Province: Implications for Conservation"

_plants, 2021, doi:10.3390/plants10112250_

Round 1
Reviewer 1 Report
It has been long time since I have read a useful and beautifully described genetic diversity paper. This paper presents an existing issue on the conservation of the endemic Dioon holmgrenii in the Mexican Pacific Coast.
The most balanced section in the manuscript was the discussion section of this MS. There is one weakness in the study which must be addressed prior to any decision.
L168-L174. Authors prose the differences in genetic diversity within the studied populations of D. holmgrenii as compared to related endemic species. Instead of just comparing the studied results with the reported ones, authors should make an actual comparison and present results after relevant statistical comparisons as written by authors as an objective of this study (see L96-98).
L170. Authors should cite the relevant table/figure here and anywhere else in the Discussion where need.
Author Response
Dear Review1
We addressed the questions made to the manuscript “Diversity and genetic structure of Dioon holmgrenii (Cycadales: Zamiaceae) in the Mexican Pacific Coast biogeographic province: Implications for conservation” as we described in the following sentences:
- The sentence on lines 96 to 98 is not part of the objective as was understood by Reviewer one. We are not able to do any statistical comparison with our data and the data obtained in other research papers as the Reviewer suggested, because we do not have the original data of those research papers. In this sentence, we want to describe to the reader that we compared our results on genetic diversity of Dioon holmgrenii with the results published on the Dioon genus by other authors, as it is common in any article even when authors do not mention it in the introduction. Thus, we decided to delete the sentence on lines 96 and 98 to avoid misunderstandings.
- We cited the tables and figures along the discussion section as the reviewer suggested.
Reviewer 2 Report
Good research work.
You travelled us to the state of Oaxaca, Mexico, emphasizing on the importance of preserving the variety and genetic structure of Dioon holmgrenii.
Good Experimental Design and Methods and high quality of presentation.
I think it would be better for the reader, the Table S3 "Characteristics of the Dioon holmgrenii populations evaluated with SSR markers" to be included in the main text (probably after table 1) and not in the Supplementary Materials.
Author Response
Dear Review2
We addressed the questions made to the manuscript “Diversity and genetic structure of Dioon holmgrenii (Cycadales: Zamiaceae) in the Mexican Pacific Coast biogeographic province: Implications for conservation” as we described in the following sentences:
- We moved the Table S3 "Characteristics of the Dioon holmgrenii populations evaluated with SSR markers" to the text as the reviewer recommended. It is now mentioned for the first time on line 405 and the Table S3 is now Table 5 in the text. We made the necessary changes in the text.
Round 2
Reviewer 1 Report
The manuscript is ok to proceed.
Author Response
Dear Academic editor,
We addressed the suggestions that you kindly gave us on the manuscript.
- We changed Ho to He as it corresponds on Line 20.
- We changed the word “distribution” to “genetic structure” on Line 26.
- We changed the word “one more diverse population” to “one most diverse population” on line 27
- We made the changes as the academic editor suggested on lines 31, 298, 303 and 306.
- We made the correction on line 191, and we deleted the plural for “genetic level”.
- We listed the species in alphabetic order in Table 4, and we made the correction on the legend Mm for Number of migrants.
- We changed the “Table S3” to Table 5 on lines 224, 228, 257, 274, 320, 346 and 355
- We changed the words “starts” to “primers” on Line 461.